# Staphylococcal Communities on Skin Are Associated with Atopic Dermatitis and Disease Severity

**DOI:** 10.3390/microorganisms9020432

**Published:** 2021-02-19

**Authors:** Sofie Marie Edslev, Caroline Meyer Olesen, Line Brok Nørreslet, Anna Cäcilia Ingham, Søren Iversen, Berit Lilje, Maja-Lisa Clausen, Jørgen Skov Jensen, Marc Stegger, Tove Agner, Paal Skytt Andersen

**Affiliations:** 1Bacteria, Parasites, and Fungi, Statens Serum Institut, Artillerivej 5, 2300 Copenhagen, Denmark; ANMC@SSI.DK (A.C.I.); sriv@ssi.dk (S.I.); BELI@SSI.dk (B.L.); JSJ@ssi.dk (J.S.J.); MTG@ssi.dk (M.S.); PSA@ssi.dk (P.S.A.); 2Department of Dermatology, Bispebjerg Hospital, Bispebjerg bakke 23, 2400 Copenhagen, Denmark; caroline.meyer.olesen@regionh.dk (C.M.O.); line.brok.noerreslet.01@regionh.dk (L.B.N.); maja-lisa.clausen.03@regionh.dk (M.-L.C.); tove.agner@regionh.dk (T.A.); 3Department of Veterinary and Animal Sciences, University of Copenhagen, Grønnegårdsvej 15, 1870 Frederiksberg, Denmark

**Keywords:** atopic dermatitis, skin microbiome, skin microbiota, *Staphylococcus*, *S. aureus*, coagulase-negative staphylococci, *tuf* amplicon sequencing, 16S rRNA amplicon sequencing, 16S rRNA qPCR

## Abstract

The skin microbiota of atopic dermatitis (AD) patients is characterized by increased *Staphylococcus aureus* colonization, which exacerbates disease symptoms and has been linked to reduced bacterial diversity. Skin bacterial communities in AD patients have mostly been described at family and genus levels, while species-level characterization has been limited. In this study, we investigated the role of the bacteria belonging to the *Staphylococcus* genus using targeted sequencing of the *tuf* gene with genus-specific primers. We compared staphylococcal communities on lesional and non-lesional skin of AD patients, as well as AD patients with healthy controls, and determined the absolute abundance of bacteria present at each site. We observed that the staphylococcal community, bacterial alpha diversity, and bacterial densities were similar on lesional and non-lesional skin, whereas AD severity was associated with significant changes in staphylococcal composition. Increased *S. aureus*, *Staphylococcus capitis*, and *Staphylococcus lugdunensis* abundances were correlated with increased severity. Conversely, *Staphylococcus hominis* abundance was negatively correlated with severity. Furthermore, *S. hominis* relative abundance was reduced on AD skin compared to healthy skin. In conclusion, various staphylococcal species appear to be important for skin health.

## 1. Introduction

Atopic dermatitis (AD) is a common inflammatory skin disease characterized by dry and itchy skin and recurring flares. The disease etiology is complex and involves interplays between immunological dysregulation and impaired skin barrier function, the most well-known genetic risk factor being loss-of-function mutations in the gene encoding the skin barrier protein filaggrin (*FLG*) [1,2]. Furthermore, AD patients are more often colonized with *Staphylococcus aureus* on their skin than healthy individuals, with a metanalysis showing a 70% colonization prevalence on lesional skin and 39% on non-lesional skin [3].

A reduction in the bacterial alpha diversity on skin, sometimes referred to as bacterial dysbiosis, has been considered to be a hallmark of the AD skin phenotype [4,5]. Yet, reduced alpha diversity may not be universal, but depend on *S. aureus* skin colonization [4,6] and the microenvironmental skin conditions, such as moisture and sebum content [7,8]. Some studies have found that alpha diversity is significantly reduced on lesional compared to non-lesional skin [9,10] whereas other have found no difference in alpha diversity between the two skin sites [5,11]. These contradictory results may suggest a continuous transition between lesional and non-lesional skin within AD patients.

The immune dysregulation and skin barrier impairment of AD patients are believed to promote skin colonization with *S. aureus*. Thus, filaggrin deficiency and reduced levels of free fatty acids in AD skin lead to increased skin pH, which, in turn, has been shown to enhance *S. aureus* growth and adherence to keratinocytes [12,13,14,15,16,17], and *FLG* loss-of-function mutations have previously been associated with increased *S. aureus* skin colonization among AD patients [18]. On the other hand, *S. aureus* skin colonization has been shown to exacerbate AD [4,19]. The elimination of *S. aureus* from clinically infected skin by the topical application of antibiotics is, therefore, one strategy used to treat AD [20]. However, this treatment approach does not have a long-lasting effect and has been shown to select for antibiotic-resistant *S. aureus* [21,22]; hence, new strategies for controlling *S. aureus* skin colonization in AD are needed. One approach has been to identify natural compounds secreted from skin commensal bacteria with bacteriocidal or bacteriostatic activity against *S. aureus*. Bacterial antagonism is more likely to occur between closely related organisms [23]; thus, other staphylococcal species may be a likely source of new antimicrobial compounds that target *S. aureus*. In accordance, some skin commensal coagulase-negative staphylococci (CoNS) are able to inhibit *S. aureus* growth, through the production of selective bacteriocins [6,24,25,26], or by inhibiting *S. aureus* virulence through the interference of the accessory gene regulator (agr) quorum-sensing system [27,28,29]. Interestingly, the application of viable bacteriocin-producing *Staphylococcus epidermidis* or *Staphylococcus hominis* on the skin of AD patients has been shown to significantly decrease the cutaneous *S. aureus* abundance 24 h after treatment when compared to non-treated patients [6], emphasizing the therapeutic potential of using skin commensal staphylococci to limit *S. aureus* colonization in AD. 

Despite their predominantly commensal relationship to their host, many CoNS are opportunistic pathogens that may infect wounds and eczematous lesions, and their role in AD may, therefore, be different to their role in healthy skin. Whereas many studies have focused on the role of *S. aureus* in AD, only a few, and often smaller, studies focusing on CoNS at species level have been published [5,19,30,31]. One of the reasons for this may be that many studies investigating the skin microbiota are based on targeted amplicon sequencing of 16S rRNA gene regions (V1–V3 or V3–V4 regions) that suffer from limited staphylococcal species discrimination of [32,33].

Here, our primary aim was to study the diversity of the staphylococcal communities on the lesional and non-lesional skin of AD patients and skin of healthy individuals using an amplicon sequencing-based method targeting the *tuf* gene. The *tuf* gene is conserved within the *Staphylococcus* genus but contains highly variable regions that can be used to discriminate between different species [32]. We also investigated staphylococcal communities on AD skin in correlation to AD severity and skin barrier dysfunction, including transepidermal water loss (TEWL) and skin pH. Secondly, we compared the overall bacterial abundance and alpha diversity on AD lesional, AD non-lesional and healthy skin using 16S rRNA (V3-V4) amplicon sequencing and qPCR.

## 2. Materials and Methods 

### 2.1. Participants

Adult AD patients from the Dermatology Outpatient Clinic at Bispebjerg Hospital (Copenhagen, Denmark) were enrolled at two time periods: November 2016–January 2017 [34] and March 2018–September 2019. Inclusion criteria were age ≥ 18 years and presence of AD according to U.K. criteria [35]. Patients were invited to participate in the study regardless of topical of systemic treatment of AD; however, any treatment within the past three months was registered (Appendix A). Disease severity was assessed by scoring AD (SCORAD), which is based on an evaluation of the skin (i.e., erythema, oedema, oozing, excoriation, lichenification, dryness and eczema extent) and a subjective assessment of symptoms (itch and sleeplessness) [36]. Common loss-of-function mutations in *FLG* (R501X, 2282del4, and R2447X) were determined using a blood test and the skin barrier function assessed by measurements of TEWL and skin pH on the volar forearm on non-lesional skin (DermaLab, Cortex Technology, Denmark) (Table 1), as described previously [34]. 

Blood donors ≥ 18 years (blood donor clinic, Hvidovre Hospital, Copenhagen, Denmark) were enrolled as healthy controls in the months of May 2019 and January 2020. Study exclusion criteria were history of skin diseases, including AD.

### 2.2. Samples

Primary samples were collected from skin using eSwabs (Copan, Brescia, Italy), and kept on dry ice for up to five hours before storing at −80 °C until further processing. For patients, lesional and non-lesional skin samples were collected depending on the site of eczema, primarily from the volar forearm and the antecubital crease (Table 1). Skin samples from healthy individuals were collected from the antecubital crease. 

*S. aureus* skin colonization was determined by culturing, by plating sample aliquots on *S. aureus*-selective plates (SaSelect, BioRad, Marnes-la-Coquette, France; or chromID, bioMérieux, France). *S. aureus* classification of colonies was validated using a *spa*-specific PCR, as previously described [37].

### 2.3. DNA Extraction, Amplicon Sequencing, and Quantitative PCR

Bacterial cells were enzymatically pre-lysed and DNA extracted using a MagNA Pure 96 purification instrument (Roche, Mannheim, Germany). The V3–V4 region of the 16S rRNA gene was selected for the analysis of complete bacterial communities [38], and the *tuf* gene was selected for analysis of staphylococcal species [32]. The selected gene regions were amplified in a two-step PCR and sequenced on a MiSeq instrument using 600-cycle MiSeq Reagent Kit v3 (Illumina Inc., San Diego, CA, USA) kits. Sequences are available at the European Nucleotide Archive under the project identification number: PRJEB42898. The absolute abundance of bacteria was estimated by qPCR using primers identical to the 16S rRNA V3–V4 primers used for amplicon sequencing. See Appendix A for details on primers and protocols.

### 2.4. Bioinformatics and Statistics

DADA2 v.1.12.1 [39] was used for the inference of amplicon sequence variants (ASVs) and taxonomic assignments using the Silva reference database v.132 [40] for 16S rRNA-derived ASVs and an inhouse reference database was used for *tuf*-derived ASVs [32]. The *tuf* reference database can be accessed here: https://github.com/ssi-dk/staphylome/tree/master/database, accessed on 18 February 2021. All statistical analyses were performed in R v.4.0.2 [41], using the R package *phyloseq* v.1.33.0 [42] for data processing, and ggplot2 v.3.3.2. [43] for the visual presentation of results. Putative contaminants were removed prior to downstream analysis using the R package *Decontam* v.3.6.1 [44].

The absolute abundance of bacteria was quantified as the total number of 16S rRNA gene copies within 1 µl of DNA eluate. For graphical visualizations, log_10_-transformed counts were used. The absolute abundance of *Staphylococcus* was estimated by combining 16S rRNA qPCR and amplicon data (relative abundance of *Staphylococcus* (amplicon data) x absolute abundance of bacteria (qPCR data)), as previously suggested in [45,46]. Absolute abundances of staphylococcal species were similarly calculated (see Appendix A for an example). 

Taxonomic counts resulting from 16S rRNA V3–V4 amplicon sequencing were agglomerated at genus level for all analyses except for alpha diversity measures, which was estimated at ASV level using the Shannon index. Differences in Shannon index, absolute abundance of bacteria, and absolute abundance of *Staphylococcus* between skin sites were tested using Wilcoxon signed-rank test and Mann–Whitney U test for paired (lesional versus non-lesional AD skin) and un-paired (AD skin versus healthy skin) samples, respectively. In addition, differences between AD lesional and non-lesional skin were tested using a linear mixed model including the skin microenvironment of the sampling area (i.e., dry, moist or sebaceous skin) as a fixed effect (Appendix A). Bacterial compositional differences between skin sites (beta diversity) were measured using Bray–Curtis dissimilarities on Hellinger-transformed counts and tested with *permutational multivariable analysis of variance* (PERMANOVA) (function adonis in R package *vegan* v.2.5.6. [47]). The determination of differentially abundant bacterial genera between skin sites was performed using *analysis of composition of microbiomes with bias correction* (ANCOM-BC) modelling, which estimates a change between test groups for each taxon using log-transformed values of absolute sequence counts [48]. The 30 most abundant genera were included in the analysis and results were corrected for multiple testing using the Benjamini-Hochberg method (i.e., controlling the false discovery rate (fdr)). Both PERMANOVA and ANCOM-BC methods allow for the adjustment of covariates and a variable containing information on the skin microenvironment of the sampling area was, therefore, included in the comparative analyses of lesional and non-lesional skin. R package *Metacoder* v.0.3.4 [49] was used for graphical presentation of the 30 most abundant bacterial genera on skin.

The *tuf*-based analysis of *Staphylococcus* was performed at species level. Staphylococcal species composition was compared between skin sites using a PERMANOVA test as described above and visualized using principal coordinates analysis (PCoA). Partitioning around medoid (PAM) clustering (R package *cluster* v.2.1.0. [50]), based on the Jensen-Shannon distance on Hellinger transformed counts, was used to group AD lesional and non-lesional skin samples into staphylococcal community clusters. Differences in bacterial Shannon diversity, absolute abundance of bacteria, SCORAD values, and TEWL measures between the four identified community clusters were tested using a Kruskal–Wallis test. If significant, a Dunn’s post hoc test was performed with correction for multiple testing using the Benjamini–Hochberg method. Differences in pH across the four community clusters were tested with an *analysis of variance* (ANOVA) test. The identification of differentially abundant species between skin sites and between AD severity groups was performed using ANCOM-BC as described for the 16S rRNA analysis. Spearman’s correlations were calculated between the absolute abundance of selected species and the following variables: Bacterial Shannon diversity, SCORAD values, skin pH measures, and TEWL measures. Results were corrected for multiple testing (Benjamini–Hochberg method) in cases where several species were included in a test. See Appendix A for an extended description. 

### 2.5. Ethical Approval

Sample and data collection was approved by the local Science Ethics Committee (Videnskabsetisk Komite, Region-Hovedstaden) (Inclusion of AD patients: H-1-2014-039 (approved 8 July 2014). Inclusion of healthy controls: H-16023435 (approved on 6 June 2016)) and by the Danish Data protection Agency. All participants provided signed informed consent.

## 3. Results

The study included 94 AD patients with active disease and 92 healthy individuals, with similar sex and age distributions as the patient cohort (Table 1). Samples were collected from both lesional and non-lesional skin areas of AD patients, primarily from the volar forearm (dry skin) and the antecubital crease (moist skin), and from the antecubital crease of healthy individuals. Since previous studies have shown that the bacterial diversity and composition can differ between anatomical skin sites [4,7,8,51], only AD skin samples collected from the antecubital crease were used in the comparative analyses with the healthy individuals. This subgroup of AD patients (*n* = 36) reflected the complete patient cohort, according to demographic and clinical characteristics (Table 1). An outline of the study design is shown in Figure 1.

### 3.1. Skin Bacterial Community Profiling 

*Staphylococcus* (Firmicutes), *Corynebacterium,* and *Micrococcus* (Actinobacteria) were the three most prevalent and abundant bacterial genera identified on the skin of both AD patients and healthy individuals (Figure 2A and Appendix A). 

#### 3.1.1. Comparison of Bacterial Communities in AD Lesional and Non-lesional Skin

When comparing the overall bacterial population on lesional and non-lesional skin from AD patients we found no differences with regards to alpha diversity (Figure 2B), or absolute abundance of bacteria (Figure 2D). Furthermore, the bacterial community structures were similar on lesional and non-lesional AD skin (R^2^ = 0.005), with none of the 30 most abundant bacterial genera, including *Staphylococcus*, being differentially abundant between the two skin sites. The absolute abundance of *Staphylococcus* was also similar on lesional and non-lesional skin (Figure 2H).

#### 3.1.2. Comparison of Bacterial Communities in AD and Skin

Since we observed similar bacterial populations of lesional and non-lesional AD skin, the samples were grouped together for the comparison of AD with healthy individuals. The absolute abundance of bacteria on the antecubital crease was significantly higher on the skin of AD patients when compared to healthy skin (median fold-change = 3.4, *p* < 0.001) (Figure 2E). Alpha diversity was reduced on AD skin (median: 2.8) compared with healthy skin (median: 3.2), although not statistically significant (*p* = 0.1) (Figure 2C). A significant difference in the composition of bacterial genera on the skin was identified between AD and healthy individuals (R^2^ = 0.02, *p* = 0.004). Differential abundance analysis indicated that *Streptococcus* was enriched and *Bacillus* and *Kocuria* abundances were reduced on AD skin compared to healthy skin (Appendix A), whereas there was no significant difference in the abundance of *Staphylococcus* on AD and healthy skin. However, the absolute abundance of *Staphylococcus* was significantly higher on AD skin (median fold change: 5.6, *p* = 0.002) (Figure 2I). 

### 3.2. Staphylococcal Communities on AD Skin

To further investigate the staphylococcal species composition on skin, we performed *Staphylococcus*-specific amplicon sequencing of the *tuf* gene.

Four distinct staphylococcal community clusters were identified based on species compositional (dis)similarities (Figure 3A, Appendix A). Community cluster 1 was characterized by high proportions of *Staphylococcus capitis* and *S. epidermidis*, community cluster 2 by *S. hominis* and *S. epidermidis* and community cluster 3 by *S. aureus*, whereas community cluster 4 was more diverse. AD skin samples belonging to community cluster 3 were further characterized by a reduced bacterial alpha diversity as compared to skin samples assigned to the other three clusters (adj. *p* < 0.001) (Figure 3C), indicating an association between *S. aureus* proportional abundance and bacterial alpha diversity. To examine this further, we performed a correlation analysis, which showed that an increase in the absolute abundance of *S. aureus* was significantly correlated with a reduced Shannon diversity within both lesional (ρ = −0.4, *p* < 0.0001) and non-lesional skin (ρ = −0.2, *p* =0.02) (Appendix A). The absolute abundance of bacteria on skin was similar in the four defined community clusters (Appendix A).

#### 3.2.1. Comparison of Staphylococcal Communities on AD Lesional and Non-Lesional Skin

There was no difference in the composition of staphylococcal species on lesional and non-lesional AD skin (R^2^ = 0.009) (Appendix A). In alignment, no significant difference in the proportion of lesional and non-lesional skin samples assigned to each community cluster was found (Figure 3A,B). Approximately half of the patients had lesional and non-lesional skin assigned to the same cluster (*n* = 46, 49%) (Figure 3B).

Differential abundance analysis showed that *S. epidermidis* was enriched in the staphylococcal community within non-lesional skin compared to lesional skin (mean fold change: 1.9, adj. *p* = 0.007), whereas the abundance of *S. aureus* was similar at the two sites (Appendix A). It may be of more clinical relevance to examine changes in absolute abundance of colonizing bacteria rather than compositional abundances within the staphylococcal community. Therefore, the absolute abundance of staphylococcal species was calculated by combining the proportional species abundance within the bacterial community with the total bacterial abundance quantified by qPCR. These data revealed that the absolute abundance of *S. aureus* was significantly increased on lesional compared to non-lesional skin (median fold change: 1.8, adj. *p* = 0.002), whereas there was no difference in the absolute abundance of *S. epidermidis* (Appendix A). 

#### 3.2.2. Staphylococcal Communities and AD Severity

AD patients with skin staphylococcal communities assigned to cluster 3 and cluster 4 had more severe AD as compared to patients with skin staphylococcal communities belonging to cluster 1 or cluster 2 (Figure 3A, Figure 4A). Differential abundance analysis showed that *S. aureus* was enriched and *S. hominis* reduced within the staphylococcal community on lesional and non-lesional skin among patients with severe AD (SCORAD > 50, *n* = 21) compared to patients with mild AD (SCORAD < 25, *n* = 25) (Figure 4B). *S. capitis* and *S. lugdunensis* were also found to be enriched on the non-lesional skin of patients with severe AD (Figure 4B). In accordance, SCORAD was positively correlated with the absolute abundance of *S. aureus* on both lesional and non-lesional skin (ρ = 0.6 and ρ = 0.5 for lesional and non-lesional skin, respectively, adj. *p*-values < 0.0001). On non-lesional skin, absolute abundances of *S. capitis* (ρ = 0.3, adj. *p* = 0.03) and *S. lugdunensis* (ρ = 0.2, adj. *p* = 0.04), were positively correlated with SCORAD, whereas *S. hominis* absolute abundance was negatively correlated (ρ = −0.3, adj. *p* = 0.02) (Appendix A). 

#### 3.2.3. Staphylococcal Communities and AD Skin Barrier Function

Changes in the skin barrier function may have an impact on *Staphylococcus* skin colonization [52], and we therefore investigated if skin pH and TEWL were associated with changes in the staphylococcal community composition. TEWL and pH were measured on non-lesional skin of the volar forearm, and thus analyses were only performed on samples collected at this skin site (*n* = 73 patients) (Figure 1). The average skin pH was higher for skin assigned to staphylococcal community cluster 3 compared to the pH of skin assigned to the three other community clusters (Figure 5A), though this was not of statistical significance. TEWL measurements were found to be similar across all four community clusters (Figure 5B). Previous studies have shown that pH affects *S. aureus* growth in vitro [16,17,53], and we therefore tested for a possible correlation between the absolute abundance of *S. aureus* and skin pH. A tendency towards increasing quantities of *S. aureus* with increasing skin pH was observed, though this was not of statistical significance (ρ = 0.2, *p* = 0.07) (Appendix A). The absolute abundance of *S. aureus* was found to be positively correlated with TEWL (ρ = 0.3, *p* = 0.01) (Appendix A). 

No significant associations were found between carriage of *FLG* loss-of-function mutations and assignment of lesional or non-lesional skin samples to the four staphylococcal community clusters.

### 3.3. Comparison of Staphylococcal Communities on AD Skin and Healthy Skin

The staphylococcal community on the skin of the antecubital crease was significantly different between AD patients and healthy individuals, with a greater difference observed between lesional and healthy skin (R^2^ = 0.13, *p* < 0.001) compared to AD non-lesional and healthy skin (R^2^ = 0.06, *p* < 0.001) (Figure 6A,B). Differential abundance analysis indicated that *S. aureus* and *S. capitis* were enriched within cutaneous staphylococcal communities of AD patients, whereas *S. hominis* and *Staphylococcus cohnii* were more abundant on the skin of healthy individuals (Appendix A). The relative abundance of *S. cohnii* within the bacterial community was, on average, below 1% and its clinical significance might, therefore, be questioned. The absolute abundance of *S. hominis* was calculated in order to adjust for the observed differences in bacterial densities on the skin of AD patients and healthy individuals (Figure 2E). Though the proportions of *S. hominis* within the bacterial community were four times higher on healthy skin compared to AD skin (*p* < 0.001) (Figure 6C), no significant difference in the absolute abundance was found between the two skin sites (Figure 6D).

## 4. Discussion

This is the first study to perform an in-depth assessment of staphylococcal species in AD patients and relate this to disease severity. The study underlines that species-level characterization of *Staphylococcus* is important and contributes significantly to the already existing knowledge on *S. aureus* and AD. 

Our results indicate that the absolute abundance of bacteria, as well as the bacterial alpha diversity and composition are similar on lesional and non-lesional skin of AD patients. Furthermore, compositional analysis suggests that the staphylococcal community is generally similar between lesional and non-lesional AD skin, whereas we found significant differences between AD and healthy skin as well as between AD severity groups. These findings indicate that AD pathogenesis has an impact on the overall skin microbiota, comprising staphylococcal species, independent of the presence of eczema. The presence of a universal bacterial community within AD skin sites has also previously been suggested [8,11], and Brandwein et al. found that increasing AD severity, rather than the presence or absence of lesions, was most important regarding changes in bacterial community within AD skin compared to healthy skin [11]. The present study of species-level characterization of *Staphylococcus* further supports this theory.

In the present study, four significantly distinct staphylococcal community clusters were identified with community cluster 1 characterized by a high proportional abundance of *S. capitis* and *S. epidermidis*, community cluster 2 by *S. hominis* and *S. epidermidis*, and community cluster 3 by *S. aureus*. Community cluster 4 was more diverse, with both *S. epidermidis*, *S. aureus*, *S. capitis*, and *S. hominis*. Bacterial alpha diversity was significantly reduced in AD skin samples belonging to community cluster 3. In accordance, an increase in the absolute abundance of *S. aureus* was correlated with a decreased bacterial alpha diversity, which supports previous reports [4,6]. Patients with lesional and non-lesional skin samples assigned to community clusters 3 and 4 had more severe AD compared to patients within community cluster 1 or 2. In accordance, increases in the absolute *S. aureus* abundance on lesional and non-lesional skin were correlated with increasing disease severity. These findings support the well-documented association between *S. aureus* skin colonization and AD severity [3,18]. The absolute abundances of *S. capitis* and *S. lugdunensis* on non-lesional skin were also associated with severe AD, whereas *S. hominis* absolute abundance was associated with a milder AD presentation. In a previous study, *S. hominis* skin colonization was found to be negatively associated with AD development during the first two years of life, whereas cutaneous *S. aureus* was predictive of AD development [54]. Although a causative relationship is uncertain, this could indicate that *S. hominis* is able to inhibit *S. aureus* skin colonization, leading to a reduced risk of AD development. In accordance, Nakatsuji et al. showed that some *S. hominis* strains produce bacteriocins acting against *S. aureus*, and that topical application of these *S. hominis* strains onto AD skin led to a significant reduction in the absolute abundance of *S. aureus* [6]. *S. lugdunensis* is also able to produce a bacteriocin that has a bactericidal effect on *S. aureus*, and it has been shown that individuals colonized with *S. lugdunensis* in the anterior nares are unlikely to be *S. aureus* nasal carriers [24]. Thus, it is surprising that we here observed that *S. aureus* and *S. lugdunensis* co-colonized the skin of patients with severe AD. A possible explanation could be nutritional differences between skin and nasal epithelium that could impact the growth and colonization properties of these species.

The composition of the staphylococcal communities on the antecubital crease of healthy individuals was significantly different from that on AD lesional and non-lesional skin, with reduced proportional abundances of *S. hominis* and *S. cohnii*, and increased abundance of *S. aureus* and *S. capitis* on AD skin. Baurecht et al. also found *S. hominis* abundance to be reduced on the antecubital crease of adult AD patients compared to healthy individuals, whereas no difference in abundance was found on dry skin of the arms [30]. *S. hominis* and *S. capitis* proportional abundances were not associated with either AD disease or disease exacerbation in children [4,19], however, these studies included less than 12 volunteers, which limits their ability to detect such associations. The importance of *S. cohnii* for human skin health is not clear, but one study has shown a negative correlation between *S. aureus* and *S. cohnii* colonization of the nasal epithelium in pigs [55]. This may indicate an antagonistic interaction between *S. aureus* and *S. cohnii*. Differences in the proportional abundance of staphylococcal species between AD patients and healthy individuals likely reflect pathophysiological abnormalities in AD skin with immune activation and decreased expression of skin barrier proteins, creating niches that facilitate different staphylococcal communities. Furthermore, one study has indicated strain-level differences in the CoNS colonizing skin of AD patients and healthy individuals [6], suggesting that it could be of relevance to investigate and compare virulence gene expression in CoNS from AD and healthy skin in future studies. Such studies could provide insight into whether *S. capitis* and *S. lugdunensis* could contribute to disease exacerbation in AD. Furthermore, staphylococcal species communities likely differ between skin anatomical areas [56], and thus it will be relevant to perform additional comparative studies of AD skin and healthy skin at other skin sites than the antecubital crease.

We found that the absolute *S. aureus* abundance on non-lesional AD skin was positively correlated with TEWL. A tendency for an association of increasing *S. aureus* abundance with increasing skin pH was also observed. A culture-based study has found that *S. aureus* skin colonization was associated with an increase in TEWL, whereas no difference in skin pH was seen between patients with *S. aureus* culture-positive or culture-negative skin swabs [52]. *In vitro* studies have shown that acidification of bacterial growth media, corresponding to the pH of healthy skin, was associated with impaired *S. aureus* growth [16,17,53] and decreased expression of *S. aureus* genes, which are important for the adherence to keratinocytes [16]. Thus, increasing skin pH might be an important mediator of enhanced *S. aureus* growth and persistent colonization in AD. Furthermore, increased TEWL reflecting impaired skin barrier function may cause enhanced penetration of *S. aureus* into the dermis, where it can trigger an immunological response [57]. At the same time, *S. aureus* may also induce skin barrier disruption by secreting proteases and cytotoxic toxins [27,58]. Since pH and TEWL were only assessed on non-lesional skin, it is unknown if the same associations would be present on lesional skin.

Some studies have shown reduced bacterial alpha diversity on AD skin compared to healthy skin [4,5,59], often denoted as microbial dysbiosis. In the present study, Shannon diversity varied between individuals, both among patients and healthy individuals, with no significant difference between AD lesional, AD non-lesional, and healthy skin. The majority of the included patients were treated with topical corticosteroids, which may have influenced the bacterial diversity of their skin [59,60]. In accordance, Kong et al. found that Shannon diversity on the antecubital crease was similar between AD patients with intermittent treatment and healthy individuals [4]. Furthermore, changes in bacterial diversity seem to be dependent on *S. aureus* colonization and abundance [4,6], which is supported by our results.

Many studies have investigated cutaneous bacterial communities in AD using compositional-based analysis, but only a few studies have examined the absolute abundance of bacteria on AD skin [9,57,59]. These studies found a significantly higher bacterial load on lesional compared to non-lesional skin, which is in contrast to the results presented in this study, where we found similar bacterial quantities on lesional and non-lesional skin. In two of these studies [9,57], lesional and non-lesional skin samples were collected from different anatomic sites, i.e., lesional samples from the antecubital crease (moist skin) and non-lesional samples from the upper arm (dry skin), which potentially could have influenced their results. In line with this, we observed a tendency towards higher bacterial density on AD non-lesional skin of moist compared to dry skin areas (*p* = 0.06, Appendix A). We also found, in agreement with previous findings [57,59] that the absolute bacterial abundance was significantly higher on AD skin compared to healthy skin, which indicates that an increase in total bacterial density is related to AD.

A major advantage of amplicon-based sequencing of bacterial target genes is the possibility of identifying fastidious bacterial species, that are difficult to culture, and thus the approach is essential when examining species diversity and microbial community composition. However, amplicon sequence data are compositional, meaning that an increase in the proportional abundance of one taxon will result in a decrease in other taxa, which is not necessarily synonymous with a change in density, as indicated by the results presented here. Thus, whereas the proportional abundance of *S. hominis* within the bacterial community was significantly reduced on AD skin compared to healthy skin, no difference in the absolute abundance of *S. hominis* was found between the two skin sites. This indicates that the reduced proportional abundance of *S. hominis* on AD skin was caused by a relative increase in other staphylococcal species, mainly *S. aureus*, and not a decrease in *S. hominis* density. Species competition within microbial communities is likely dependent on the absolute abundance of species [61]. Furthermore, the absolute abundance of a species can, in some cases, also be important for regulation of bacterial virulence, as was observed with the population density-modulated agr quorum sensing pathway, which regulates the expression of important *S. aureus* virulence genes [62,63]. Thus, investigating changes in absolute rather than relative abundance is likely of clinical relevance.

Our study has some limitations. One relates to the inconsistency of skin anatomical sites being sampled from AD patients and healthy individuals. Consequently, only a minor subset of AD skin samples was included in the comparative analyses with healthy skin, limiting the statistical power. Furthermore, the use of topical and systemic treatments among some of the patients may have influenced the results. However, this reflects the situation in a real-life clinical setting. Another limitation is that the absolute abundance of species was estimated using 16S rRNA qPCR, without correcting for copy number variations in the gene across species. Thus, our findings should be confirmed in future studies quantifying staphylococcal species abundance by qPCR targeting *Staphylococcus*-specific single-copy genes. However, the quantification of absolute abundance, and not solely proportional abundances, constitutes a significant strength of the study. Another major strength of the study is the use of *tuf*-based amplicon sequencing leading to reliable staphylococci classification at species level and, finally, the high number of included patients compared to previous studies in this field of research, which allows for stronger statistical power. 

In conclusion, we found that increased abundance of *S. aureus*, *S. capitis,* and *S. lugdunensis* are positively correlated with increasing severity, whereas *S. hominis* abundance showed a negative correlation, and that *S. hominis* proportional abundance was reduced on AD skin compared to healthy skin. This indicates that multiple staphylococcal species, and not just *S. aureus*, have an important role in AD regarding chronic disease and exacerbations. Furthermore, our data suggest that the pathophysiological abnormalities in AD skin have a universal impact on the skin microbiota, that is independent of the presence of current eczema status. 

## Figures and Tables

**Figure 1 microorganisms-09-00432-f001:**
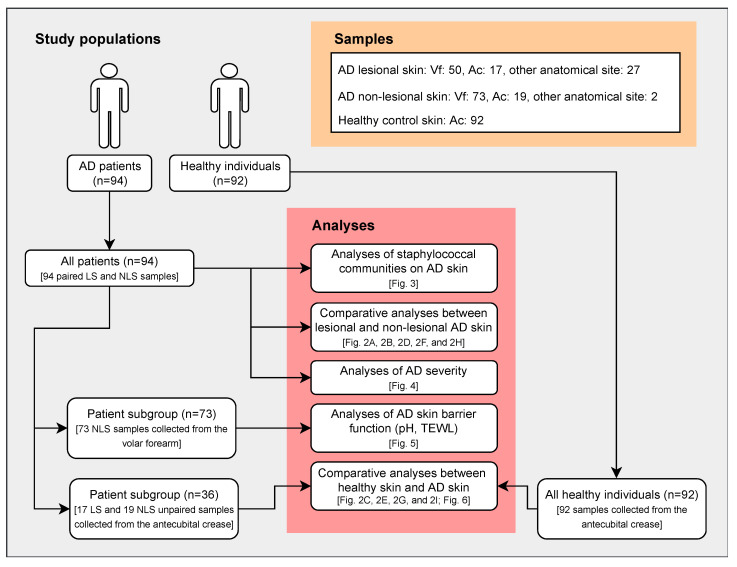
Study design. Illustration showing study subjects, samples, and performed analyses. Skin samples from healthy individuals were all collected at the antecubital crease. Consequently, only AD skin samples collected from the antecubital crease were included in comparative analyses between AD patients and healthy individuals. TEWL and pH were measured on the volar forearm on non-lesional AD skin, and only samples collected from this site were, therefore, included in analyses of skin barrier function. Abbreviations: AD: atopic dermatitis; LS: lesional skin; NLS: non-lesional skin; TEWL: transepidermal water loss; Vf: volar forearm; Ac: antecubital crease.

**Figure 2 microorganisms-09-00432-f002:**
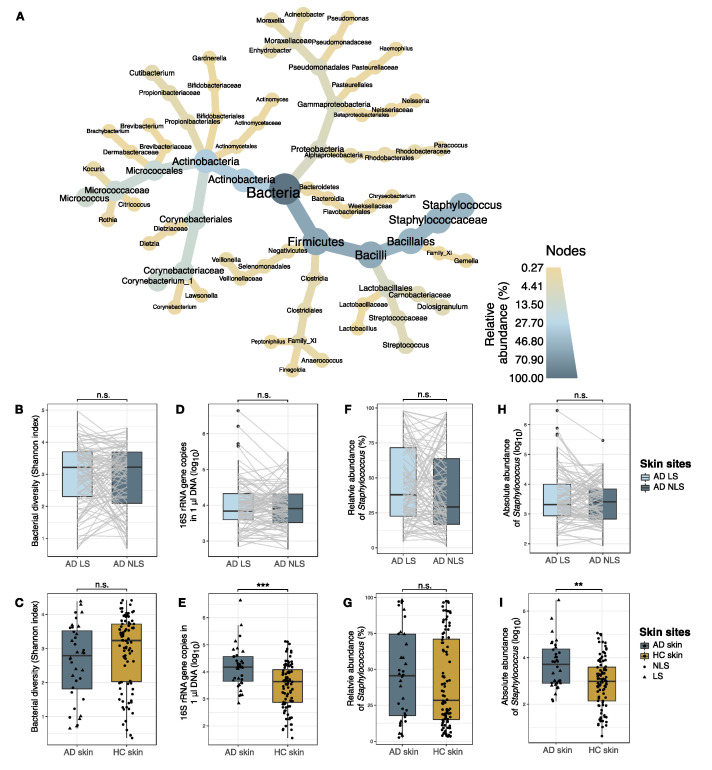
Skin bacterial communities. (**A**) Taxonomic illustration of the 30 most abundant genera on AD skin. Color and size of nodes and leaves correspond to the average relative abundance of taxa. AD lesional and non-lesional skin samples were grouped together. (**B**,**C**). Bacterial alpha diversity measured by Shannon index. (**D**,**E**) Absolute abundance of bacteria, measured as 16S rRNA gene copies within 1 µl DNA sample eluate. (**F**,**G**) Relative abundance (%) of *Staphylococcus* genus. Differences between skin sites were tested using ANCOM-BC modelling on log transformed sequence counts. (**H**,**I**) Absolute abundance of *Staphylococcus*, calculated by combining 16S rRNA qPCR and amplicon sequence data. In **B**, **D**, **F**, and **H**, lesional and non-lesional skin of AD patients (*n* = 94) were compared. Samples collected from the same patient are indicated by grey lines. In **C**, **E**, **G**, and **I**, skin samples collected from AD patients (*n* = 36) and healthy individuals (*n* = 92) were compared. Here, AD lesional (triangles) and non-lesional (dots) skin samples were grouped together as AD skin. Boxplots represent the median and interquartile range (IQR) with whiskers extending to the minimum/maximum value, but no longer than 1.5xIQR. Asterisks indicate statistical significance: ** *p* < 0.01, *** *p* < 0.001, *n*.s.: not significant. Abbreviations: AD: atopic dermatitis; LS: lesional skin; NLS: non-lesional skin; HC: healthy control skin.

**Figure 3 microorganisms-09-00432-f003:**
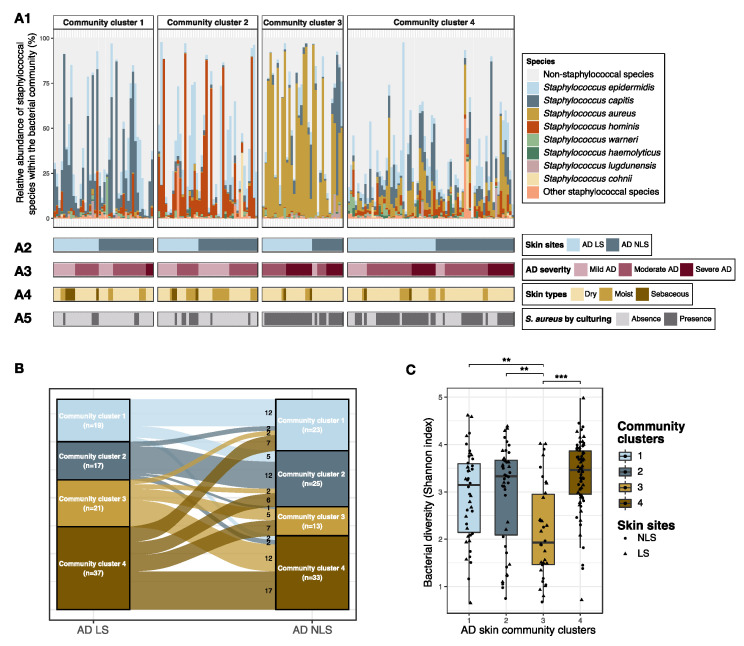
Staphylococcal communities on AD skin. Lesional and non-lesional skin from AD patients (*n* = 94) were grouped into four distinct community clusters based on compositional similarities of staphylococcal species. (**A1**) Bar plot showing the relative abundances of the eight most abundant staphylococcal species within the bacterial community. (**A2**) Tile plot showing whether samples were collected from lesional or non-lesional skin. (**A3**) Tile plot showing the AD severity of patients from whom the samples were collected. Mild AD: SCORAD < 25; Moderate AD: SCORAD 25–50; and Severe AD: SCORAD > 50. (**A4**) Tile plot showing whether samples were collected from dry, moist, or sebaceous skin areas. (**A5**) Tile plot showing *S. aureus* colonization status, defined by culturing. (**B**) Sankey diagram for intra-individual comparison of staphylococcal community cluster assignment of lesional and non-lesional skin. (**C**) Comparison of skin bacterial alpha diversity, measured by Shannon index, across the four defined staphylococcal community clusters. Lesional and non-lesional skin samples were visualized as triangles and dots, respectively, and boxplots represent the median and interquartile range (IQR) with whiskers extending to the minimum/maximum value, but no longer than 1.5 × IQR. Asterisks indicate statistical significance: ** adj. *p* < 0.01, *** adj. *p* < 0.001. Abbreviations: AD: atopic dermatitis; LS: lesional skin; NLS: non-lesional skin.

**Figure 4 microorganisms-09-00432-f004:**
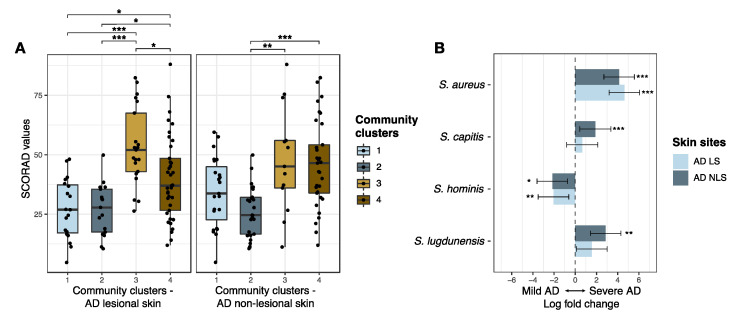
AD severity and cutaneous staphylococcal communities**.** (**A**) AD severity, assessed by SCORAD, and its association to staphylococcal community cluster assignments of lesional and non-lesional skin of AD patients (*n* = 94). Boxplots represent the median and interquartile range (IQR) with whiskers extending to the minimum/maximum value, but no longer than 1.5xIQR. (**B**) Differential abundant staphylococcal species on lesional and non-lesional skin from patients with mild AD (SCORAD < 25, *n* = 25) as compared to patients with severe AD (SCORAD > 50, *n* = 21). Bars represent the ANCOM-BC estimated log fold change in species abundance between compared groups (effect size) and error bars, with the 95% confidence interval. Log fold change refers to the natural logarithm. Species with a fdr-adjusted *p*-value < 0.05 were considered significant and are shown in the plot. Asterisks indicate statistical significance: * adj. *p* < 0.05, ** adj. *p* < 0.01, *** adj. *p* < 0.001. Abbreviations: AD: atopic dermatitis; LS: lesional skin; NLS: non-lesional skin; SCORAD: Scoring Atopic Dermatitis.

**Figure 5 microorganisms-09-00432-f005:**
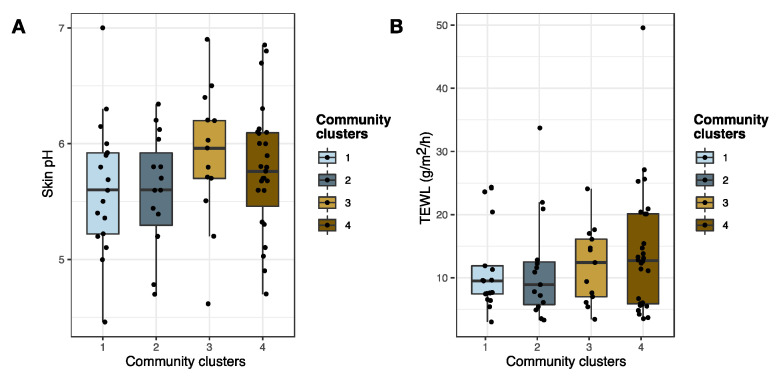
Skin barrier function (pH and TEWL) and cutaneous staphylococcal communities**.** Skin pH (**A**) and TEWL (**B**) and their association to staphylococcal community cluster assignment of non-lesional AD skin (*n* = 73). Boxplots represent the median and interquartile range (IQR) with whiskers extending to the minimum/maximum value, but no longer than 1.5xIQR. Abbreviations: AD: atopic dermatitis; TEWL: transepidermal water loss.

**Figure 6 microorganisms-09-00432-f006:**
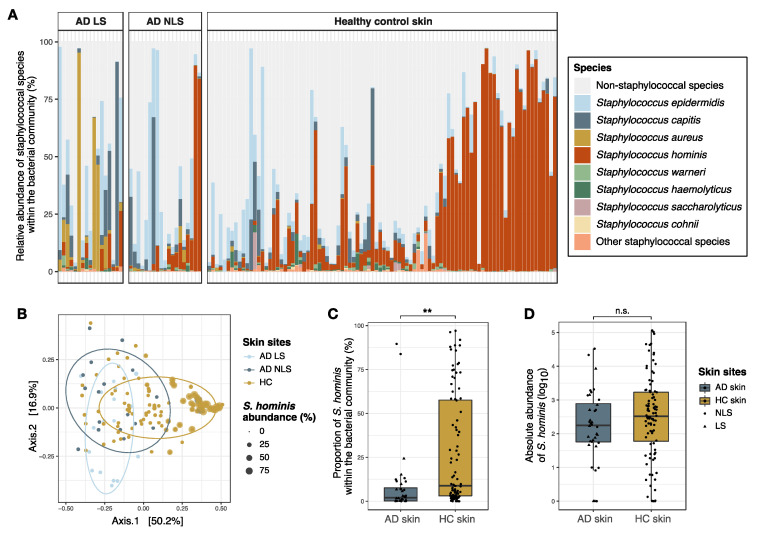
Staphylococcal communities on the antecubital crease of AD patients and healthy individuals**.**
*(***A**) Bar plot showing the relative abundances of the eight most abundant staphylococcal species within the bacterial community of AD lesional skin (*n* = 17), AD non-lesional skin (*n* = 19), and healthy control skin (*n* = 92). (**B)** PCoA plot showing the compositional differences in staphylococcal communities on AD skin and healthy control skin. Point sizes are scaled based on the relative abundance of *S. hominis* within the bacterial community. Of note, two AD non-lesional skin samples with high *S. hominis* abundance cluster together with healthy control skin on the right side of the plot. (**C)** The relative abundance of *S. hominis* within the cutaneous bacterial community. (**D**) The absolute abundance of *S. hominis* on skin, which was calculated by combining proportional amplicon sequence data with 16S rRNA qPCR data. Thus, the absolute abundance is equivalent to the number of *S. hominis* 16S rRNA gene copies within 1µl DNA eluate (log10 transformed counts). Box plots represent the median and interquartile range (IQR) with whiskers extending to the minimum/maximum value, but no longer than 1.5 × IQR, and lesional and non-lesional skin samples are visualized as triangles and dots, respectively. Asterisks indicate a significant difference: ** *p* < 0.01, *n*.s.: not significant. Abbreviations: AD: atopic dermatitis; LS; lesional skin; NLS: non-lesional skin; HC skin: healthy control skin.

**Table 1 microorganisms-09-00432-t001:** Description of study populations.

Covariates		AD Patients (*n* = 94)	Sub-Group of AD Patients (*n* = 36)	Healthy Individuals (*n* = 92)
**Demographic data**				
Sex	Female:Male ratio	44:50	18:18	47:45
Age (years)	Median (range)	38 (18–71)	36 (21–71)	40 (18–66)
**Clinical data**				
Other atopic diseases	Asthma	47 (50%)	19 (53%)	1 (1%)
	Allergic rhinitis	67 (71%)	26 (72%)	13 (14%)
	Type 1 allergy	79 (83%)	29 (81%)	14 (15%)
AD severity (SCORAD) ^1^	Median (range)	38 (5–88)	34.0 (4.7–88)	NR
	Mild:Moderate:Severe	25:48:21	14:17:5	NR
*FLG* loss-of-function mutations ^2^	Mutations	34 (36%)	12 (33%)	NE
	WT	50 (53%)	22 (61%)	NE
Skin pH ^3^	Median (range)	5.7 (4.5–7.0)	5.7 (4.7–6.8)	NE
TEWL ^3^ (g/m^2^/h)	Median (range)	10 (3–50)	10 (4–34)	NE
***S. aureus* culture data**				
Colonization of LS		58 (62%)	20 (56%)	NR
Colonization of NLS		32 (34%)	10 (28%)	5 (6%)
Colonization of Nares		61 (66%)	20 (56%)	31 (34%)
**Sample location**				
LS sample location ^4^	Vf:Ac:Other	50:17:27	0:17:0	NR
NLS sample location ^4^	Vf:Ac:Other	73:19:2	0:19:0	0:92:0
Microenvironment of LS	Dry:Moist:Sebaceous	64:18:12	0:17:0	NR
Microenvironment of NLS	Dry:Moist:Sebaceous	75:19:0	0:19:0	0:92:0

^1^ Mild disease: SCORAD < 25; Moderate disease: SCORAD 25–50; Severe disease: SCORAD > 50. ^2^ Mutations: R501X, 2282del4, and/or R2447X. Wild type (WT) represents absence of any of the three mutations. Ten patients were not tested for *FLG* mutations. ^3^ skin pH and TEWL were measured on non-lesional skin of the volar forearm. ^4^ Vf: volar forearm. Ac: antecubital crease. “Other” includes the back, cheeks, chest, hand, leg, neck, popliteal crease, and thigh. Abbreviations: AD: atopic dermatitis; LS: lesional skin; NLS: non-lesional skin; *FLG*: filaggrin gene; TEWL: transepidermal water loss; NR: Not relevant; NE: Not examined.

## Data Availability

All sequences are available through the European Nucleotide Archive (project number: PRJEB42898). R scripts and phyloseq objects are available at: https://github.com/ssi-dk/AD_staphylome_project, accessed on 18 February 2021. Due to national data protection regulations regarding personally identifiable information, only a limited number of variables are included in the sample data file.

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
