# Peer review of "Staphylococcal Communities on Skin Are Associated with Atopic Dermatitis and Disease Severity"

_microorganisms, 2021, doi:10.3390/microorganisms9020432_

Round 1
Reviewer 1 Report
Edslev et al. has investigated the staphylococcal communities on lesional and non-lesional skin of AD patients and on skin of healthy individuals. This is very interesting study which was well designed with moderate high number of samples. Several up to date methods were applied to analyze the data. Staphycococcal communities on skin were not only compared by AD and healthy people but also between various sites and different severe levels of AD patients. This data provided an important information of skin microbiota at staphylococcal species associated with AD disease. Additionally, the paper was clear written with a fruitful discussion.
Questions:
Recently, Ahle et al. (PMID: 32718033) has showed that the third most abundant skin member was Staphylococcus saccharolyticus. However, your study gave no information about this species. Did you blast your data to this new species, I am wondering whether you can detect this human skin species?
In Figure 4B, are there any alterations of S. epidermidis abundance between the comparison of patients with severe AD and mild AD?
Reviewer 2 Report
This manuscript has identified different species of Staphylococcus genus involved in atopic dermatitis by using tuf gene. They compared isolates on lesioned and non-lesioned skin of AD patients along with controls. Authors showed that S. aureus, Staphylococcus capitis, and Staphylococcus lugdunensis abundances correlated with increased severity whereas abundances of S. hominis correlates with decreased severity. Though not completely novel, it has very useful data for AD bacterial communities and their interactions.
I have following minor comments.
- Method section needs more description on how the samples been processed. Line 108 says eSwabs and grown on selective media. Selective media for different sub species of Staphylococcus or only for Staphylococcus aureus?
- Since all Staphylococcal species mentioned in these manuscripts are varied widely in terms of growth kinetics, optimal growth condition, doubling time and also production of bacteriocins, it may favor growth of one species over other in artificial growth media. Hence, this study would have been more informative, if performed parallelly with direct PCR for tuf gene versus culture from eSwab.
